# Sarcopenia as a predictor of survival in patients undergoing bland transarterial embolization for unresectable hepatocellular carcinoma

Ezio Lanza[1]* , Chiara Masetti[2] , Gaia Messana[3] , Riccardo Muglia[1,2,3], Nicola Pugliese[2,3], Roberto Ceriani[2], Ana Lleo de Nalda[2,3], Lorenza Rimassa[3,4], Guido Torzilli[5], Dario Poretti[1], Felice D'Antuono[1], Letterio Salvatore Politi[3,4,5,6], Vittorio Pedicini[1‡], Alessio Aghemo[2,3‡], on behalf of the Humanitas HCC Multidisciplinary Group[¶]

1 Division of Interventional Radiology, Department of Radiology, Humanitas Research Hospital IRCCS, Rozzano, Italy, 2 Division of Internal Medicine and Hepatology, Department of Gastroenterology, Humanitas Research Hospital IRCCS, Rozzano, Italy, 3 Department of Biomedical Sciences, Humanitas University, Pieve Emanuele, Milan, Italy, 4 Medical Oncology and Hematology Unit, Humanitas Cancer Center, Humanitas Clinical and Research Center-IRCCS, Rozzano, Milan, Italy, 5 Division of Hepatobiliary & General Surgery, Department of Surgery, Humanitas Clinical and Research Center IRCCS, Rozzano, Italy, 6 Department of Radiology, Humanitas Clinical and Research Center IRCCS, Rozzano, Italy

☯ These authors contributed equally to this work.
‡ VP and AA are Joint Senior Authors.
¶ Membership of the Humanitas HCC Multidisciplinary Group is provided in the Acknowledgments.
* eziolanza@gmail.com

**Data Availability Statement:** All relevant data are within the paper and its Supporting Information files.

## Abstract

Sarcopenia has been associated with lower overall survival in patients with cirrhosis and hepatocellular carcinoma (HCC) undergoing surgical resection, TACE, TARE, or transplantation. This monocentric study evaluated the prognostic significance of sarcopenia in patients affected by HCC who received bland transarterial embolization (TAE) therapy, by analyzing its impact on survival and treatment-related complications. All consecutive patients who underwent the 1st TAE between March 1st 2011 and July 1st 2019 in our Institution were retrospectively studied. To evaluate sarcopenia, the skeletal muscle index (SMI) was calculated by normalizing the cross-sectional muscle area at the level of L3 on an abdominal CT scan prior to embolization (cm2) by patient height (m2). SMI cut-off values for sarcopenia were considered ≤ 39 cm2/m2 for women and ≤55 cm2/m2 for men. Data about age, gender, body mass index (BMI), underlying liver disease, liver function, MELD score, Child-Pugh score, multifocal disease, performance status, previous interventions, length of stay (LOS), complications after the procedure, readmission rate within 30 days, survival time from TAE and total number and type of TAE received following the first procedure were collected. From 2011 to 2019, 142 consecutive patients underwent 305 TAEs. Observation time ranged from 1.4 to 100.5 months (median 20.1 SD = 22). Sarcopenia at baseline was present in 121 (85%) patients. Overall 87 (61.2%) patients died during follow-up with survival rates at 1-, 2-, 3-, 4-, and 5-year of 71%, 41%, 22%, 16% and 11% respectively. After multivariate analysis sarcopenia (HR = 2.22, p = 0.046), previous ablation/

**Funding:** The author(s) received no specific funding for this work.

**Competing interests:** NO authors have competing interests.

resection (HR = 0.51, p = 0.005) and multifocal disease (HR = 1.84, p = 0.02) were associated with reduced survival. Sarcopenia did not influence the safety of TAE in terms of LOS (2 days vs 1.5 days, p = 0.2), early complications rate (8% vs 5%, p = 0.5) and readmission rate within 30 days (7% vs 5%, p = 0.74). Sarcopenia, estimated by the L3SMI method, is an emerging prognostic factor in patients with HCC undergoing bland TAE therapy as it is associated with increased mortality, without impairing the safety of the locoregional treatment. Measures to ameliorate the SMI, such as nutritional support and physical exercise, should be evaluated in clinical trials for HCC patients receiving liver embolization to determine their impact on overall survival.

## Introduction

Patients affected by Hepatocellular Carcinoma (HCC) are often unfit for curative surgery due to advanced disease at diagnosis, portal hypertension, reduced liver function, multinodular disease and concomitant comorbidities [1,2]. As a consequence, according to the Barcelona Clinic Liver Cancer (BCLC) staging system, they are frequently referred for intra-arterial therapies including transarterial embolization (TAE) for local disease control, before resorting to systemic treatments when they are no longer responsive to locoregional treatments or when they progress to BCLC stage C [3,4]. Among the different techniques, bland transarterial embolization (TAE) has been shown to improve patient survival and tumor response with similar efficacy when compared to transarterial chemoembolization (TACE) or transarterial radioembolization (TARE) [4,5]. Relying only on the ischemic effect of small microparticles, TAE does not cause the side effects known to TACE and it is less expensive and less invasive than TARE [6]. Survival rates following TAE are extremely heterogeneous as patients who undergo TAE vary significantly in terms of liver function, age and concomitant comorbidities. On top of these differences, HCC burden (number of nodules and vascular invasion) and its molecular features play a significant role in the survival post-TAE [7]. Sarcopenia, defined as loss of skeletal muscle mass, has emerged as a negative prognostic factor in patients with HCC undergoing surgical resection, TACE, Sorafenib therapy and liver transplantation, as well as in patients with other solid tumors [8–11]. However, data regarding the impact of sarcopenia in patients receiving TAE monotherapy are lacking.

The objectives of this study were to assess the impact of sarcopenia estimated by skeletal muscle index (SMI) on overall survival and the incidence of postprocedural complications in patients with HCC who underwent TAE in a single center (Humanitas Research Hospital IRCCS).

## Materials and methods

### Design of the study

This retrospective analysis aimed to evaluate if sarcopenia could be a predictor of overall survival (OS) in a cohort of patients who received TAE for HCC. The study was approved by the Ethics Committee of the Humanitas Research Hospital IRCCS and was conducted in accordance with the ethical standards laid down in the Declaration of Helsinki. Informed written consent was obtained before every treatment from all patients.

Indication to TAE was given by a multidisciplinary team comprising liver surgeons, oncologists, interventional radiologists, radiotherapists, and hepatologists [12]. All patients

underwent a full disease staging before being referred to TAE and they were followed up with CT imaging 1 month after TAE. In the event of disease recurrence or partial response to treatment, the cases were re-discussed; otherwise, a 3 months follow-up interval was established.

Patients with a diagnosis of HCC who underwent the first TAE in our Institute between March 1st, 2011 and July 1st, 2019 and who had an abdominal CT scan performed up to two months before the procedure were included; those who had only an abdominal MRI were excluded due to the impossibility of calculating the SMI. The end date of the observation period was set to January 17th, 2020.

All TAEs were performed without the use of intra-arterial chemotherapeutic drugs. Technical variations included microparticles TAE (P-TAE), microparticles plus cyanoacrylate glue TAE (G-TAE) and Lipiodol TAE (L-TAE), as previously reported by our group [6]. All procedures were performed by two senior operators and two junior interventional radiologists with at least three years' experience in an angiographic suite (V5000 Philips Medical System, Amsterdam, The Netherlands), equipped with cone-beam CT (Siemens Artis-Zee, Munich, Germany).

We collected the following data for each patient: age, gender, BMI, underlying liver disease (HCV, HBV, HDV, alcohol abuse, NASH), liver function tests (ALT, AST, AFP, PLTs, INR, albumin, bilirubin), MELD score, Child-Pugh score, number of HCC nodules, performance status, previous interventions (surgery, stereotactic body radiation therapy -SBRT- or thermal ablation), LOS, early complications rate, readmission rate within 30 days and survival time from TAE. We also considered the total number and type of TAE received following the first procedure.

## Assessment of sarcopenia

Sarcopenia was assessed using the L3-SMI method [13,14]. On the latest available abdominal CT scan performed before the first TAE procedure, we segmented the lean muscle mass at an axial plane located at the level of the third lumbar vertebra (L3), in which both transverse processes were visible (Fig 1).

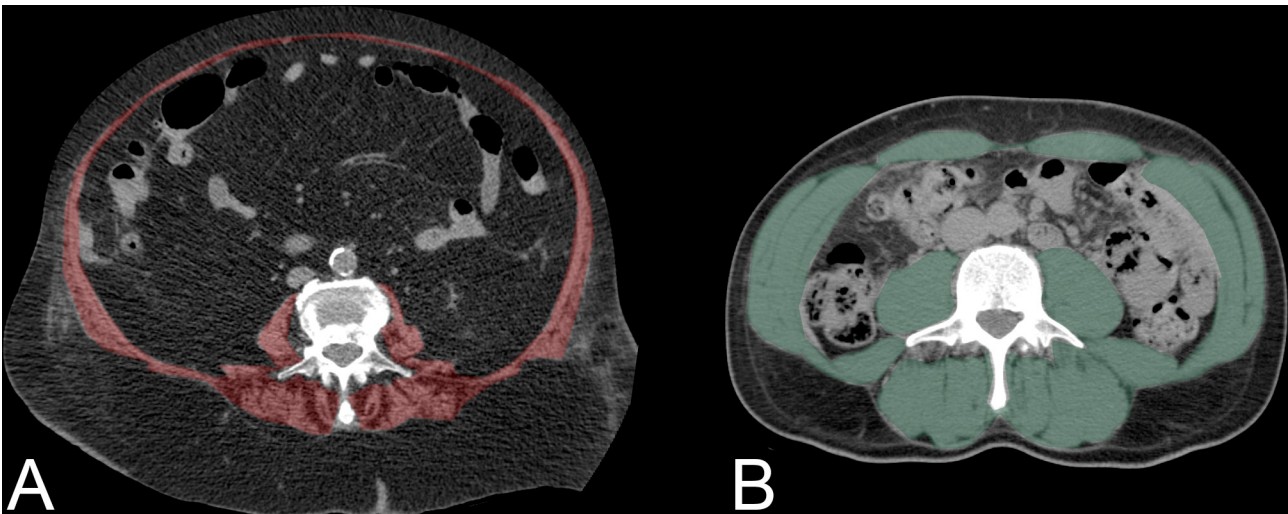

**Fig 1. SMI measurement at the level of the L3 lumbar vertebra. A**. The red area highlights the abdominal muscles in a female patient aged 84 years, highly sarcopenic (SMI = 21,1) who survived only 21 days from the first TAE **B**. The green area highlights the same muscles in a highly muscular male patient aged 60 years (SMI = 64,6, survival 45 months and alive at the end of the study).

This included psoas, rectus and transversus abdominis, internal and external oblique, quadratus lumborum, longissimus thoracis, iliocostalis lumborum and spinalis thoracis muscles. Images were analyzed using dedicated software (Medstation, Exprivia, Italy) to calculate the total cross-sectional muscular area from pixels in the density range of -29 to +150 Hounsfield Units (HUs). Manual corrections to subtract areas with HU values -30 were performed in order to avoid overestimation of muscular tissue or erroneous inclusion of adjacent visceral, subcutaneous or intramuscular adipose tissue.

Then, we obtained the SMI normalizing the cross-sectional muscle area (cm2) by patient height (m2) [14], SMI cut-off values for sarcopenia were set at $\leq$ 39 cm2/m2 for women and $\leq$55 cm2/m2 for men [15].

## Statistical analysis

Stata 13 (StataCorp LP, Texas, USA) was used for all calculations. Survival analysis was performed by E. L. who has five years of experience in medical statistics. Multivariate analyses were conducted using Cox regression. Death was considered the primary outcome and was ascertained using the regional registry. T-test and Wilcoxon Rank-sum test were used to analyze differences between sarcopenic and non-sarcopenic patients for specific instances.

Predictors of survival were estimated using univariate analyses and log-rank tests for categorical variables. We tested the following potential interactions: TAE-related complications and number of treatments; previous treatments and number of HCC nodules. Kaplan-Meier curves were also generated (Fig 2).

## Results

### Descriptive data of the study population

From 2011 to 2019, we included 142 patients (32 F, 110 M). Patient characteristics are shown in Table 1.

At baseline, the mean age was 73 year old (median = 75, SD = 9.5, range 40–88).

The most common liver diseases were HCV infection in 65 cases (46%) and alcohol abuse in 33 (23%); 9 patients (6%) had an unknown cause of liver disease. One-hundred-six patients (75%) had a multifocal disease (mean number of nodules = 2 range 1–3) with the largest tumor measuring averagely 34mm (SD = 22; median = 28mm, range 7–115).

The Eastern Cooperative Oncology Group (ECOG) performance status was 0 in 57 patients (40%), 1 in 68 patients (48%), 2 in 16 patients (11%) and in one patient it could not be assessed.

The MELD score was in average 10 (SD = 3; median = 9, range 6–29); Child-Pugh score was class A in 93 patients (81.5%), class B in 20 (17.5%) and class C in 1 patient (1%).

BCLC was very early stage (0) in seven patients (5%); early stage (A) in 31 patients (21.4%); intermediate stage (B) in 81 patients (57%); advanced stage (C) in 21 patients (14.8%); terminal stage (D) in one patient (0.7%) and in one patient it could not be assessed.

Eighty-six patients (61%) had not undergone previous treatment for HCC while fifty-six patients (39%) had a history of previous percutaneous ablation or liver resection. Overall, 305 TAEs were performed (average 2.2, SD = 1.4): in detail, 143 G-TAEs (47%), 123 P-TAEs (40%) and 39 L-TAEs (13%) (Table 2).

The average BMI was 25.8 (SD = 4.9; median 25.5, range 16.7–53); the mean SMI was 41 (SD = 8.9; median 41, range 21.1–64.8).

Sarcopenia, defined as SMI cut-off values $\leq$ 39 cm2/m2 for women and $\leq$55 cm2/m2 for men, was present in 121 (85%) patients. The prevalence of sarcopenia was not significantly different between patients with single or multiple HCC nodules.

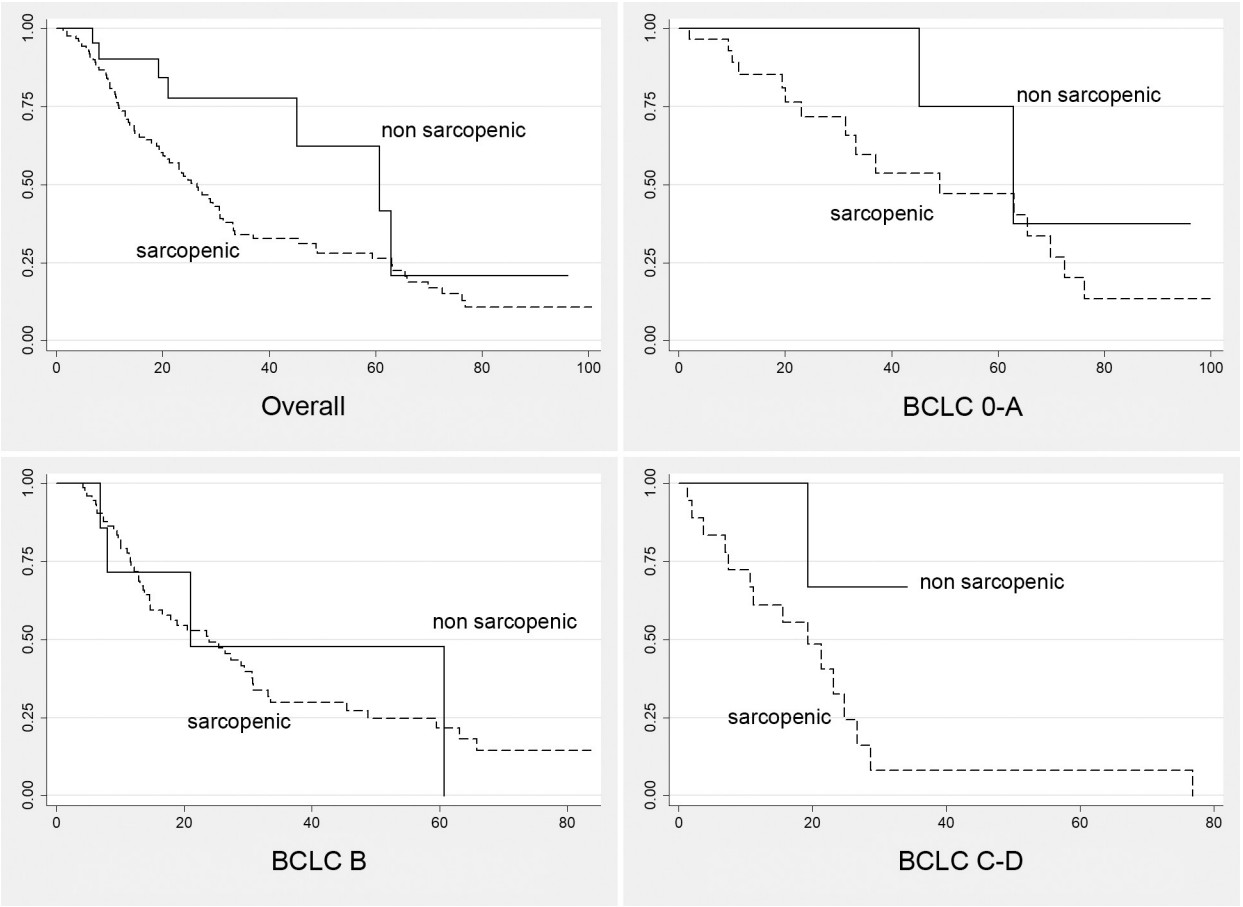

**Fig 2. Kaplan meier curves.** Survival curves of patients with sarcopenia (dashed line, SMI ≤ 39 cm2/m2 for women and ≤55 cm2/m2 for men) versus non-sarcopenic patients (continuous line). Curves were generated also according to HCC stage (BCLC 0-A, B and C-D).

**Table 1. Comparison between sarcopenic and non-sarcopenic patients according to different variables.**

| Variable (mean) | Sarcopenic | Non-sarcopenic | p-value |
|---|---|---|---|
| Age | 73 (40–88) | 73 (48–84) | 0.76 |
| Unifocal disease | 29 | 7 | 0.36 |
| Multifocal disease | 92 | 14 | |
| Largest lesion size | 34mm (7–115) | 29mm (10–86) | 0.3 |
| Gender female | 24 | 8 | 0.06 |
| HCV+ | 58 | 7 | 0.21 |
| HBV+ | 7 | 0 | 0.25 |
| Alcohol abuse | 29 | 4 | 0.62 |
| NASH | 14 | 7 | 0.01 |
| Diabetes | 40 | 8 | 0.65 |
| Child-Pugh A | 93 | 16 | 0.26 |
| Child-Pugh B | 20 | 1 | 0.18 |
| Child-Pugh C | 1 | 0 | 0.67 |

**Table 2. Number and types of TAE procedures performed.**

| TOT Patients | n˚ of TAE for each patient | G-TAE | P-TAE | L-TAE |
|---|---|---|---|---|
| 59 | 1 | 25 | 22 | 12 |
| 41 | 2 | 43 | 31 | 8 |
| 19 | 3 | 31 | 21 | 5 |
| 13 | 4 | 19 | 24 | 9 |
| 7 | 5 | 14 | 18 | 3 |
| 2 | 6 | 7 | 4 | 1 |
| 1 | 8 | 4 | 3 | 1 |
| 142 | - | 143 | 123 | 39 |

## Impact of sarcopenia on the safety of TAE

Mean LOS after TAE was 2 days (SD = 1.7, range 1–15). After the procedure, 11 patients (8%) had early complications (6 had digestive bleeding, 3 had sepsis and 2 had liver decompensation) and 9 (6%) had unplanned readmission within 30 days. No statistically significant differences were found between sarcopenic and non-sarcopenic patients concerning the following: LOS (2 days vs 1.5 days, p = 0.2), early complications rate (8% vs 5%, p = 0.5) and rate of readmission within 30 days (7% vs 5%, p = 0.74).

## Impact of sarcopenia on survival

Patients were followed up regularly for a mean of 27 months (1.4–100.5 months: median 20.1; SD = 22).

Eighty-seven (61.2%) patients died during the follow-up period; with survival rates at 1-, 2-, 3-, 4-, and 5-year being 71%, 41%, 22%, 16% and 11% respectively. (Table 3, Fig 3).

When testing OS prediction, the univariate analysis (Table 4) resulted in the inclusion of the following variables: being sarcopenic, performance status, multifocal disease, history of previous surgical or ablative treatments, MELD score and patient weight. After multivariate analysis (Table 5), the following were predictors of lower OS: sarcopenia (HR = 2.22, p = 0.046), previous ablation/resection (HR = 0.52, p = 0.005) and multifocal disease (HR = 1.84, p = 0.02). The potential interactions proved non-significant and were not included in the final model.

## Discussion

The European Working Group on Sarcopenia in Older People (EWGSOP) defines sarcopenia as "a syndrome characterized by progressive and generalized loss of skeletal muscle mass and strength with a risk of adverse outcomes such as physical disability, poor quality of life and

**Table 3. Patients overall survival and subgroup analysis.**

| | n | Overall Survival | | | | |
|---|---|---|---|---|---|---|
| | | 1yr | 2yrs | 3yrs | 4yrs | 5yrs |
| PATIENTS | | 101 | 58 | 32 | 23 | 16 |
| | | 71% | 41% | 22% | 16% | 11% |
| Sarcopenic | 121 | 83 | 47 | 26 | 20 | 14 |
| | | 69% | 39% | 21% | 17% | 12% |
| Non-sarcopenic | 21 | 18 | 11 | 6 | 3 | 2 |
| | | 86% | 52% | 29% | 14% | 10% |

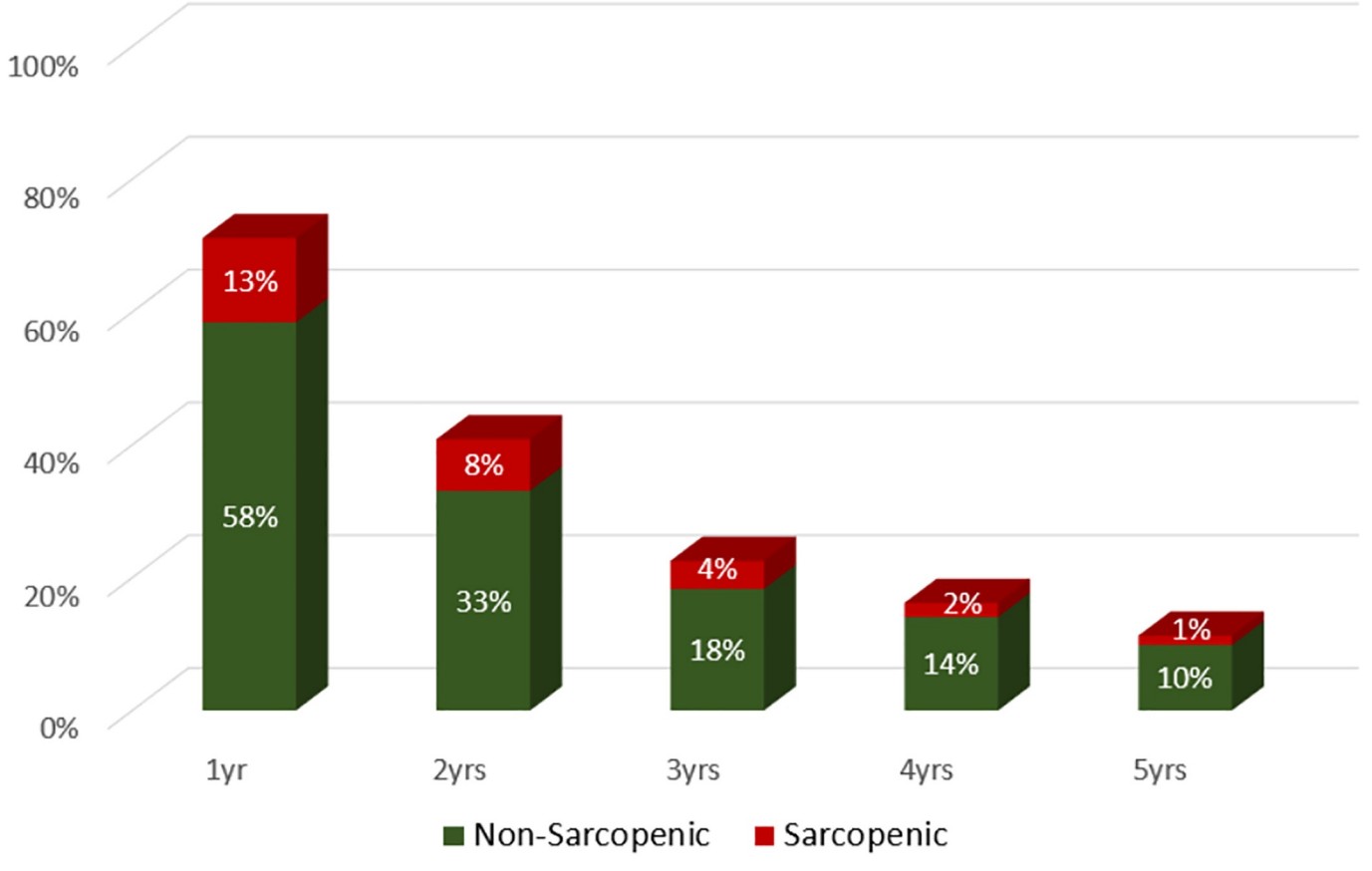

**Fig 3. Survival rates of the population grouped by the presence of sarcopenia.**

**Table 4. Variables tested in the univariate survival analysis (n = 87).**

| Variable | Y | N | p-value |
|---|---|---|---|
| Sarcopenia | 80 | 7 | 0.03* |
| Performance status = 0 | 31 | 56 | 0.03* |
| Performance status > 2 | 12 | 75 | 0.05* |
| Multifocal disease | 67 | 20 | 0.01* |
| Previous treatments | 30 | 57 | 0.03* |
| MELD score 0–9 | 40 | 47 | 0.59 |
| MELD score 10–19 | 44 | 43 | 0.64 |
| MELD score >19 | 0 | 87 | 0.57 |
| Weight | mean = 73 kg SD = 16 | | 0.599 |
| Age at diagnosis | M = 74 (40–88) F = 76 (42–88) | | 0.886 |

*included in the multivariate analysis

P <0.2 allows for inclusion in the multivariate.

**Table 5. Variables tested in the multivariate analysis.**

| Variable | HR | SD | z | p | C.I. 95% | |
|---|---|---|---|---|---|---|
| Sarcopenia * | 2.22 | 0.88 | 2.14 | 0.046 | 1.01 | 4.86 |
| Performance status = 0 | 0.73 | 0.17 | -1.26 | 0.21 | 0.46 | 1.18 |
| Performance status > 2 | 1.67 | 0.6 | 1.79 | 0.07 | 0.95 | 3.50 |
| Previous treatments * | 0.51 | 0.12 | -2.79 | 0.005 | 0.32 | 0.81 |
| Multifocal disease * | 1.84 | 0.48 | 2.31 | 0.02 | 1.09 | 3.10 |

*predictor of survival

P < 0.05 is considered significant.

death" [16]. Sarcopenia has been recently associated with increased mortality in patients with HCC or cirrhosis, regardless of the treatment received [17]. Particularly, its prognostic value has been analyzed in patients with HCC undergoing surgical resection, TACE therapy, Sorafenib therapy, or transplantation. In our Institution, however, TAE is preferred over TACE in patients with unresectable HCC as it provides similar survival rates without the side effects related to arterial chemotherapy [6,18]. Briefly, the aim of TAE therapy, as opposed to TACE, is to deploy the embolizing agent from a distal stance, in close proximity to the tumor, to achieve the maximum ischemic effect and promote necrosis of tumoral cells.

Of note, the advantage of adding chemotherapy to bland embolization is a controversial topic [18]. In this retrospective survival analysis conducted in consecutive patients who underwent TAE at our center, we observed that sarcopenia is a negative prognostic factor also for HCC patients receiving TAE monotherapy, highlighting its role as a potential determinant for patient stratification and selection of candidates. Other factors affected OS: first, a history of surgical resection or percutaneous ablation prior to TAE, which may have implied a long-standing disease; second, the presence of multiple as opposed to a single unresectable nodules (Fig 4), indicator of a greater disease burden, which may account for a poorer life expectancy [19].

Survival rates in non-sarcopenic patients were slightly lower at 4- and 5-year compared to sarcopenic patients (14% vs 17%; 10% vs 12%). However, this did not impact the statistical significance of our results, hence no correlation with sarcopenia may be assumed.

Sarcopenia was not associated with higher rates of post-procedural complications nor impaired safety of TAE, thus indicating that the increased mortality in sarcopenic patients was not the consequence of technical complications of TAE. This finding has important clinical implications as it highlights that pre-TAE sarcopenia should not be considered an exclusion criterion for the procedure; on the contrary, it should prompt an effort to increase SMI aimed to OS improvement. Among the strategies studied for the management and prevention of sarcopenia, only physical exercise [20,21] has shown significant positive effects. A potential role for Creatine and Leucine supplementation has been suggested but remains yet to be proven [22,23]. Thus, further well-designed ad-hoc studies are needed [24], even if it might be difficult to conduct them in patients with advanced liver disease, older age and significant extrahepatic comorbidities who are usually the subgroups receiving TAE. Most importantly, given that sarcopenia was not the only factor associated with mortality and that the other variables correlated to reduced survival are mostly non-modifiable and intrinsically associated with late HCC diagnosis, it remains to be proven that the exclusive correction of sarcopenia will have a positive effect in patients undergoing TAE.

There were some limitations to this study. Firstly, due to its retrospective nature, we could not acquire information on the patient's daily activities and diet, which may have influenced the prevalence of sarcopenia. Secondly, data about the specific cause of patients' death during the follow-up period were not covered by our study; thus external factors may have influenced

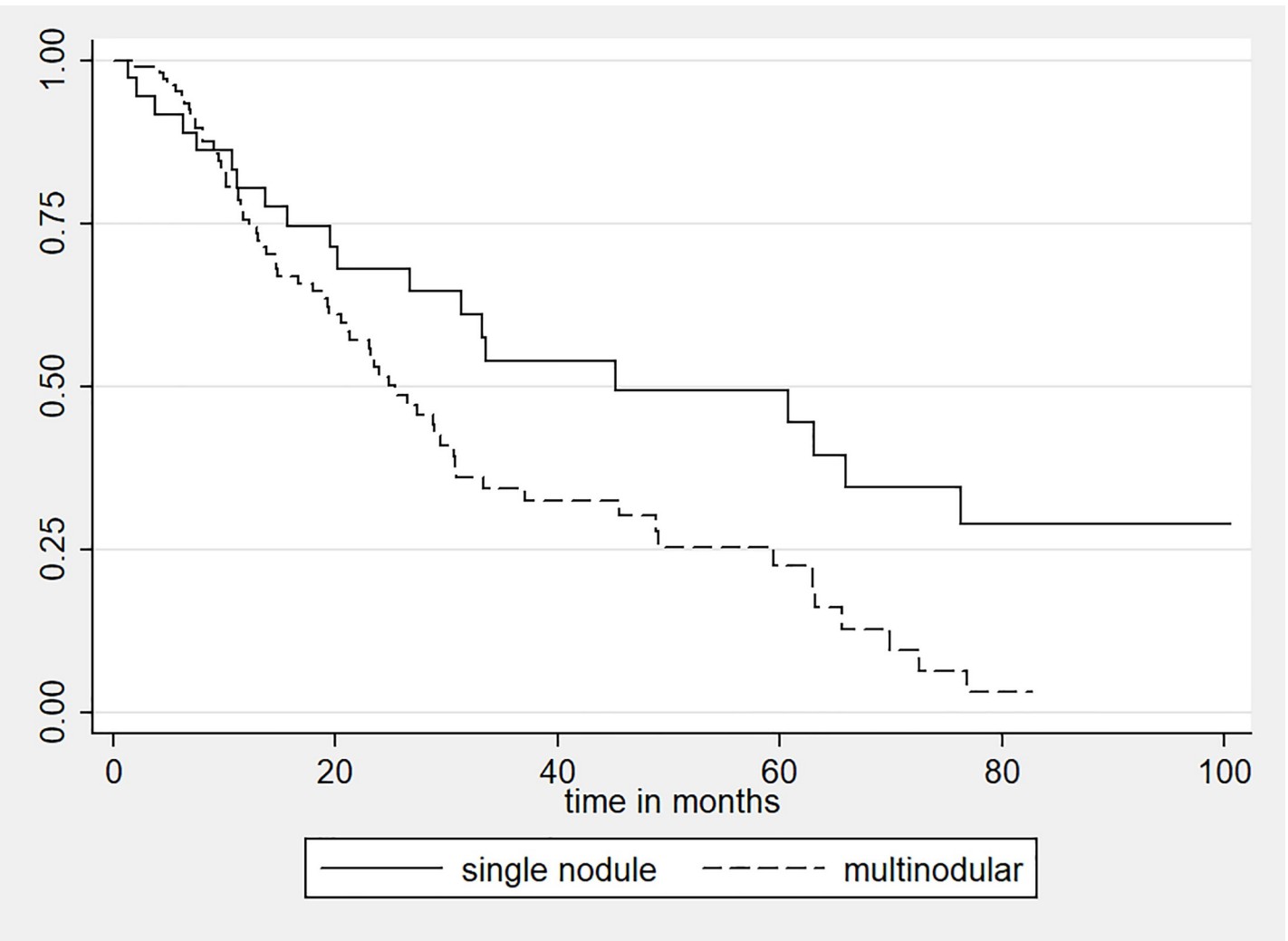

**Fig 4. Survival curves of HCC patients with a single nodule (continuous line) versus patients with a multinodular disease (dashed line).**

the OS. Thirdly, only a small percentage of patients were not sarcopenic (15%), reflecting the fact that sarcopenia is common in patients affected by advanced-stage HCC who are usually the ideal candidates for TAE, but reducing the comparability of the two groups. Also, the treatment history differed in the cohort we studied, thus potentially influencing the OS. However, this should be considered representative of HCC patients who undergo embolization, as they are frequently diagnosed at different stages, with TAE often being their first therapeutic option before systemic therapy.

Noteworthy, the strengths of this study are the large cohort managed by the same group of physicians, and the exclusion of patients who underwent any other HCC treatment during the observation window, limiting the confounding effect on OS of other therapeutic interventions.

## Conclusions

Sarcopenia, estimated by the L3SMI method, is an emerging prognostic factor in patients with HCC undergoing bland TAE therapy as it is associated with increased mortality, without impairing the safety of the locoregional treatment. Measures to ameliorate the SMI, such as

nutritional support and physical exercise, should be evaluated in clinical trials for HCC patients receiving liver embolization to determine their impact on overall survival.

## Supporting information

**S1 File. STATA database containing all data used for the statistical analysis.** (ZIP)

## Acknowledgments

Humanitas HCC Multidisciplinary Group

  **Lead Author:** Prof. Guido Torzilli (guido.torzilli@humanitas.it)

  **Members**:

| | | |
|---|---|---|
| Alessio Aghemo | Ciro Franzese | Vittorio Quagliuolo |
| Luca Balzarini | Roberto Gabbiadini | Alessandro Repici |
| Ilaria Bianchi | Silvia Garlaschi | Lorenza Rimassa |
| Isabella Bolengo | Elena Generali | Marcello Rodari |
| Silvia Bozzarelli | Nicolò Gennaro | Massimo Roncalli |
| Savino Bruno | Luigi Laghi | Matteo Sacchi |
| Michele Carvello | Ezio Lanza | Marta Scorsetti |
| Roberto Ceriani | Ana Lleo De Nalda | Carlo Selmi |
| Silvia Chiola | Egesta Lopci | Valeria Smiroldo |
| Arturo Chiti | Fabio Romano Lutman | Martina Sollini |
| Matteo Maria Cimino | Paola Magnoni | Antonino Spinelli |
| Francesca Colapietro | Alberto Malesci | Maurizio Tommasini |
| Massimo Colombo | Arianna Marinello | Guido Torzilli |
| Tiziana Comito | Chiara Masetti | Simona Verlingieri |
| Nadia Cordua | Francesca Meda | Edoardo Vespa |
| Guido Costa | Marco Montorsi | Luca Viganò |
| Giovanni Covini | Riccardo Muglia | Giuseppe Ferrillo |
| Luca Cozzaglio | Angela Palmisano | Fabio Procopio |
| Antonio D'Alessio | Vittorio Pedicini | |
| Federica D'Antonio | Nicola Personeni | |
| Daniele Del Fabbro | Francesca Piccoli | |
| Luca Di Tommaso | Laura Poliani | |
| Angelo Dipasquale | Dario Poretti | |
| Roberto Doci | Tiziana Pressiani | |
| Matteo Donadon | Maria Prete | |

## Author Contributions

**Conceptualization:** Ezio Lanza, Chiara Masetti, Lorenza Rimassa, Alessio Aghemo.

**Data curation:** Ezio Lanza, Chiara Masetti, Nicola Pugliese.

**Formal analysis:** Ezio Lanza.

**Investigation:** Ezio Lanza, Roberto Ceriani, Dario Poretti, Felice D'Antuono.

**Project administration:** Vittorio Pedicini, Alessio Aghemo.

**Resources:** Guido Torzilli, Vittorio Pedicini, Alessio Aghemo.

**Software:** Ezio Lanza.

**Supervision:** Ana Lleo de Nalda, Lorenza Rimassa, Guido Torzilli, Dario Poretti, Letterio Salvatore Politi.

**Validation:** Ezio Lanza, Lorenza Rimassa, Guido Torzilli, Letterio Salvatore Politi, Vittorio Pedicini, Alessio Aghemo.

**Visualization:** Ezio Lanza, Vittorio Pedicini.

**Writing – original draft:** Ezio Lanza, Gaia Messana.

**Writing – review & editing:** Ezio Lanza, Gaia Messana, Riccardo Muglia, Lorenza Rimassa, Alessio Aghemo.

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
