## [Decision Letter · Decision Letter 0]

9 Apr 2020

PONE-D-20-08174

Sarcopenia as a predictor of survival in patients undergoing bland transarterial embolization for unresectable Hepatocellular Carcinoma

PLOS ONE

Dear Dr. Ezio Lanza,

Thank you for submitting your manuscript to PLOS ONE. After careful consideration, we feel that it has merit but does not fully meet PLOS ONE’s publication criteria as it currently stands. Therefore, we invite you to submit a revised version of the manuscript that addresses the points raised during the review process.

We would appreciate receiving your revised manuscript within 60 days. To enhance the reproducibility of your results, we recommend that if applicable you deposit your laboratory protocols in protocols.io, where a protocol can be assigned its own identifier (DOI) such that it can be cited independently in the future. For instructions see: http://journals.plos.org/plosone/s/submission-guidelines#loc-laboratory-protocols

We look forward to receiving your revised manuscript.

Kind regards,

Gianfranco D. Alpini

Academic Editor

PLOS ONE

Journal Requirements:

2. Please amend your current ethics statement to address the following concerns: Please state whether informed consent was written or verbal. If written consent was not obtained, please describe how you recorded/documented participant consent, and if the ethics committees/IRBs approved this consent procedure.

3. One of the noted authors is a group or consortium "Humanitas HCC Multidisciplinary Group" In addition to naming the author group, please list the individual authors and affiliations within this group in the acknowledgments section of your manuscript. Please also indicate clearly a lead author for this group along with a contact email address.

Reviewers' comments:

Reviewer's Responses to Questions

**Comments to the Author**

1. Is the manuscript technically sound, and do the data support the conclusions?

Reviewer #1: Yes

Reviewer #2: Partly

2. Has the statistical analysis been performed appropriately and rigorously? 

Reviewer #1: Yes

Reviewer #2: Yes

3. Have the authors made all data underlying the findings in their manuscript fully available?

Reviewer #1: Yes

Reviewer #2: Yes

4. Is the manuscript presented in an intelligible fashion and written in standard English?

Reviewer #1: Yes

Reviewer #2: Yes

5. Review Comments to the Author

Reviewer #1: This manuscript is a mono-centric study by the Humanitas HCC Multidisciplinary Group in Milan (Italy). It is a retrospective study aiming to evaluate the prognostic significance of sarcopenia in patients affected by HCC who received bland transarterial embolization (TAE) therapy. The manuscript is interesting and well-written. I have only few minor points.

- In table 1, authors could include also alcohol abuse and NAFLD diagnosis as variables.

- Line 179. Authors stated “No specific liver disease was associated with sarcopenia (Table 4); non-NASH liver disease showed the highest incidence (p < 0.01)”. However, table 4 reports variables tested in the multivariate analysis.

Reviewer #2: The authors provide insight on an important factor determining HCC TAE outcome, however there are some issues that must be addressed prior to publication:

1. Ranges must be included along with the mean in the various tables provided with this manuscript. Additionally, subcategories of underlying diseases of sarcopenic and non-sarcopenic patients must be added in additional rows in the appropriate table.

2. There must a table with the TAE data that was listed in the manuscript.

3. Results should be rewritten for a more cohesive presentation of the data collected.

4. Correlation analysis should be performed between sarcopenia and the following:

a. Previous treatment and HCC nodule #

b. TAE complication and TAE #

5. Additionally, there should be analysis that separates the OS of sarcopenic/non-sarcopenic patients for the various stages. This data will likely be weaker due to the lower N, but it will help lend evidence to the idea that sarcopenia may influence OS and early intervention will help depending on HCC stage.

6. The authors should discuss the lower OS in non-sarcopenic patients in years 3 and 4 compared to sarcopenic patients.

6. PLOS authors have the option to publish the peer review history of their article (what does this mean?). If published, this will include your full peer review and any attached files.

Reviewer #1: No

Reviewer #2: No

---

## [Author Response · Author response to Decision Letter 0]

10 Apr 2020

Response to Reviewers 

Dear Professor Alpini, 

Thank you for giving us the opportunity to submit a revised draft of the manuscript “Sarcopenia as a predictor of survival in patients undergoing bland transarterial embolization for unresectable hepatocellular carcinoma”. We appreciate the constructive suggestions and the attentive comments you and the reviewers have made. We have incorporated most of them; changes are marked-up within the file “Revised Manuscript with Track Changes”. 

Our response follows (Editor’s and reviewers’ comments are in italics). 

Journal requirements

1. Please ensure that your manuscript meets PLOS ONE's style requirements, including those for file naming. The PLOS ONE style templates can be found at http://www.journals.plos.org/plosone/s/file?id=wjVg/PLOSOne_formatting_sample_main_body.pdf
http://www.journals.plos.org/plosone/s/file?id=ba62/PLOSOne_formatting_sample_title_authors_affiliations.pdf

Reply: The corrections have been made.

2. Please amend your current ethics statement to address the following concerns: Please state whether informed consent was written or verbal. If written consent was not obtained, please describe how you recorded/documented participant consent, and if the ethics committees/IRBs approved this consent procedure.

Reply: We specified in the text that informed consent was written. Thank you for pointing this out.

3. One of the noted authors is a group or consortium "Humanitas HCC Multidisciplinary Group" In addition to naming the author group, please list the individual authors and affiliations within this group in the acknowledgements section of your manuscript. Please also indicate clearly a lead author for this group along with a contact email address.

Reply: We added the members list of Humanitas HCC Multidisciplinary Group in the Acknowledgments section. 

Reply: We added captions for our Supporting Information.

Review Comments to the Author

Reviewer #1: This manuscript is a mono-centric study by the Humanitas HCC Multidisciplinary Group in Milan (Italy). It is a retrospective study aiming to evaluate the prognostic significance of sarcopenia in patients affected by HCC who received bland transarterial embolization (TAE) therapy. The manuscript is interesting and well-written. I have only few minor points.

- In table 1, authors could include also alcohol abuse and NAFLD diagnosis as variables.

- Line 179. Authors stated “No specific liver disease was associated with sarcopenia (Table 4); non-NASH liver disease showed the highest incidence (p < 0.01)”. However, table 4 reports variables tested in the multivariate analysis.

Reply: We included in Table 1 the variables “Alcohol abuse” and “NASH” as suggested. We removed the reported sentence because there was an error, thank you for pointing this out. 

Reviewer #2: The authors provide insight on an important factor determining HCC TAE outcome, however there are some issues that must be addressed prior to publication:

1. Ranges must be included along with the mean in the various tables provided with this manuscript. Additionally, subcategories of underlying diseases of sarcopenic and non-sarcopenic patients must be added in additional rows in the appropriate table.

Reply: Ranges were included in the tables for continuous variables. HBV+, NASH and alcohol abuse were added as variables in Table 1. 

2. There must a table with the TAE data that was listed in the manuscript.

Reply: We created a table with the TAE data, specifying the number and types of TAE procedures performed (Table 2). 

3. Results should be rewritten for a more cohesive presentation of the data collected.

Reply: We reorganized the data included in the Results in three main paragraphs: “Descriptive data of the study population”, “Impact of sarcopenia on the safety of TAE”, “Impact of sarcopenia on survival”. We believe that details are necessary for the deep comprehension of the study and should be included in the text. Instead, to have a general schematic view, Tables should be used.

4. Correlation analysis should be performed between sarcopenia and the following: a. Previous treatment and HCC nodule # b. TAE complication and TAE # 

Reply: We have tested the interactions and both proved non-significant (previous treatments#HCC, p = 0.070; TAE complication#number of TAE, p = 0.65). We have added these data in the “Materials and Methods” and “Results” section. The sentences read:

“We tested the following potential interactions: TAE-related complications and number of treatments; previous treatments and number of HCC nodules.” (Materials and methods)

“The potential interactions proved non-significant and were not included in the final model.” (Results)

5. Additionally, there should be analysis that separates the OS of sarcopenic/non-sarcopenic patients for the various stages. This data will likely be weaker due to the lower N, but it will help lend evidence to the idea that sarcopenia may influence OS and early intervention will help depending on HCC stage.

Reply: We generated Kaplan-Meier curves according to disease stage (BCLC), distinguishing sarcopenic patients from non-sarcopenic patients. They are incorporated in Fig 2.

6. The authors should discuss the lower OS in non-sarcopenic patients in years 3 and 4 compared to sarcopenic patients.

Reply: The lower OS at 4- and 5-years in the non-sarcopenic group with respect to the sarcopenic group was examined in the Discussion.

---

## [Decision Letter · Decision Letter 1]

14 Apr 2020

Sarcopenia as a predictor of survival in patients undergoing bland transarterial embolization for unresectable Hepatocellular Carcinoma

PONE-D-20-08174R1

Dear Dr. Ezio Lanza,

We are pleased to inform you that your manuscript has been judged scientifically suitable for publication and will be formally accepted for publication once it complies with all outstanding technical requirements.

With kind regards,

Gianfranco D. Alpini

Academic Editor

PLOS ONE

Additional Editor Comments (optional):

Reviewers' comments:

Reviewer's Responses to Questions

**Comments to the Author**

1. If the authors have adequately addressed your comments raised in a previous round of review and you feel that this manuscript is now acceptable for publication, you may indicate that here to bypass the “Comments to the Author” section, enter your conflict of interest statement in the “Confidential to Editor” section, and submit your "Accept" recommendation.

Reviewer #2: All comments have been addressed

2. Is the manuscript technically sound, and do the data support the conclusions?

Reviewer #2: Yes

3. Has the statistical analysis been performed appropriately and rigorously? 

Reviewer #2: Yes

4. Have the authors made all data underlying the findings in their manuscript fully available?

Reviewer #2: Yes

5. Is the manuscript presented in an intelligible fashion and written in standard English?

Reviewer #2: Yes

6. Review Comments to the Author

Reviewer #2: The authors have either addressed the reviewer critiques or explained as to why text should not be changed. There are no further comments or recommendations.

7. PLOS authors have the option to publish the peer review history of their article (what does this mean?). If published, this will include your full peer review and any attached files.

Reviewer #2: No

---

## [Editor Report · Acceptance letter]

24 Apr 2020

PONE-D-20-08174R1 

Sarcopenia as a predictor of survival in patients undergoing bland transarterial embolization for unresectable Hepatocellular Carcinoma 

Dear Dr. Lanza:

I am pleased to inform you that your manuscript has been deemed suitable for publication in PLOS ONE. Congratulations! Your manuscript is now with our production department. 

With kind regards,

on behalf of

Dr. Gianfranco D. Alpini 

Academic Editor

PLOS ONE